# Computational and *In silico* study of novel fungicides against combating root rot, gray mold, fusarium wilt, and cereal rust

Mollah Naimuzzaman[1,2☯], Md Mahabub Hasan[1,2☯], Ajoy Kumer[3*], Abu Yousuf Hossin[1], Mohammad Harun-Ur-Rashid[3], Swapan Kumar Roy[1], Abu Noman Faruq Ahmmed[4], Jamal Uddin[5]

1 College of Agricultural Sciences, IUBAT—International University of Business Agriculture and Technology, Dhaka, Bangladesh, 2 IUBAT Innovation and Entrepreneurship center, IUBAT—International University of Business Agriculture and Technology, Dhaka, Bangladesh, 3 Department of Chemistry, College of Arts and Sciences, IUBAT—International University of Business Agriculture and Technology, Dhaka, Bangladesh, 4 Department of Plant Pathology, Sher-e-Bangla Agricultural University, Dhaka, Bangladesh, 5 Department of Natural Sciences, Center for Nanotechnology, Coppin State University, Baltimore, Maryland, United States of America

☯ These authors contributed equally to this work.
* kumarajoy.cu@gmail.com

## Abstract

The exploration of potential candidates for fungicides against four fungal proteins that cause some vital plant diseases, namely *Phytophthora capsici*, *Botrytis cinerea*, *Fusarium oxysporum* f. sp. *lycopersici*, and *Puccinia graminis* f. sp. *tritici*, was conducted using in silico, molecular docking simulations, and molecular dynamic (MD) simulation for selecting the nature of binding affinity with actives sites of proteins. First of all, the DFT was employed to optimize the molecular geometry, and get the prepared optimized ligand. From the DFT data, the chemical descriptors were calculated. Next, two docking tools, such as AutoDock by PyRx and Molecular Docking by Glide from the Schrödinger suite, were used to convey the docking score, and ligand protein interactions against four main proteases, for instance 7VEM, 8H6Q, 8EBB, and 7XDS having name of pathogens: *Phytophthora capsici*, *Botrytis cinerea*, *Fusarium oxysporum* f. sp. *lycopersici*, and *Puccinia graminis* f. sp. *tritici*, respectively. In case of auto dock from PyRx, the fungicides L01, L03, L04, L13, L14, L17, L18, and L19 demonstrated significantly higher affinities for binding to the four fungal pathogens. Surprisingly, it is conveyed that the L03 illustrated the highest binding score against three of 7VEM, 8EBB, and 7XDS proteins and L09 is highest for 8H6Q. However, MD was performed to check the validation and calculation the docking procedure and stability of the protein ligand docked complex accounting of RMSD, RMSF, SASA, Radius of gyration (Rg), Protein secondary structure elements (SSE), Ramachandran plot which confirm that the stability of docked complex is so high, and number of calculating the hydrogen bonds is more than good enough, as a result it is concluded the docking procedure is valid. Finally, Difenoconazole (L03) has been considered as the most promising antifungal drug evaluated from the studies.

**Data availability statement:** All data are in the manuscript and/or supporting information files.

**Funding:** The author(s) received no specific funding for this work.

**Competing interests:** The authors have declared that no competing interests exist

## Introduction

The broad and negative impact of plant pathogenic fungi on agriculture, ecology, and food security around the world highlights the urgent need for the development of novel fungicides. These fungal diseases are responsible for considerable crop productivity and quality losses, endangering farmers' livelihoods as well as the global food supply chain [1]. Furthermore, plant diseases induced by fungi contribute to economic instability in agricultural businesses [2], increase pesticide dependency [3], and harm the environment owing to excessive chemical use [4]. The emergence of plant pathogenic fungus exacerbates this problem. Many fungus species have evolved resistance to current fungicides, making conventional chemical treatments ineffective over time. This resistance not only reduces the efficacy of disease control measures, but also requires greater dosages or the use of several fungicides, raising the danger of environmental pollution and harm to non-target organisms [5].

Developing novel fungicides is therefore critical to addressing these difficulties. These innovative methods must be tailored to target fungal diseases with greater precision, reduce environmental impact, and overcome resistance mechanisms. Advanced technologies, including as nanotechnology, bio-based fungicides, and molecular biology techniques, are laying the groundwork for the next generation of fungicides [6]. These techniques allow researchers to design targeted treatments that disrupt fungal life cycles while causing minimum harm to beneficial microbes and the surrounding ecosystem. Furthermore, novel fungicides are critical to guaranteeing sustainable farming operations. By incorporating them into crop protection measures, farmers can minimize their reliance on standard chemical treatments and adopt more environmentally friendly practices. This transition is critical for preserving biodiversity, protecting soil health, and sustaining long-term productivity in agricultural systems [7]. However, the compelling necessity for innovative fungicides arises from the profound impact of plant pathogenic fungi [8], such as *Phytophthora capsici*(PDB ID: 7vem) [9], *Botrytis cinerea* (PDB ID: 8H6Q) [10], *Fusarium oxysporum* f. sp. *lycopersici* (PDB ID: 8EBB) [11], and *Puccinia graminis* f. sp. tritici (PDB ID: 7XDS) [12], on global agriculture. These fungal pathogens are responsible for notable crop damages and economic losses [13], with *Fusarium oxysporum* causing annual economic losses worldwide due to its detrimental effects on tomato production [14]. The escalating resistance to traditional fungicides and their environmental repercussions further emphasize the critical necessity for alternative approaches [15]. Recent progress in computational methodologies, such as molecular docking [16], Molecular Dynamics (MD) simulations [17], Density Functional Theory (DFT) [18], as well as in silico techniques, present promising avenues for the discovery and enhancement of novel fungicidal targets.

Fungal infections significantly impact food security and cause widespread human suffering [19]. The Bengal Famine, aggravated by the fungal pathogen *Cochliobolus miyabeanus* inducing brown spot disease in rice [20], led to an estimated 2 to 3 million fatalities due to starvation and malnutrition [21,22]. Likewise, the Irish Potato Famine was instigated by *Phytophthora infestans*, the causative agent of late blight in potatoes [23]. *Phytophthora capsici*, an oomycete of great concern, induces root rot [24], stem blight, and fruit rot in various crops, notably peppers [25], particularly thriving in warm and moist environments [26]. The pathogen is disseminated through soil and water, utilizing zoospores that travel via soil water films and contaminated irrigation systems [27]. *Phytophthora capsici*, leading to significant economic repercussions amounting to millions of dollars annually, is especially evident in the United States [28].

The proteins 7VEM, 8H6Q, 8EBB, and 7XDS, derived from various devastating plant pathogens, represent critical targets for computational drug discovery aimed at combating agricultural diseases. The protein 7VEM, identified as NADPH-assisted quinone oxidoreductase from *Phytophthora capsici*, plays a crucial role in maintaining redox balance essential for

the pathogen's survival and virulence, making it a prime target for inhibitor design. Solved at 2.39 Å resolution with an R-work value of 0.167, its structural fidelity and lack of mutations ensure reliable insights for virtual screening and molecular docking. Similarly, the Class I sesquiterpene synthase BCBOT2 (apo) from *Botrytis cinerea* (PDB ID: 8H6Q), resolved at 2.00 Å with R-free and R-work values of 0.195 and 0.168 respectively, is vital for sesquiterpene biosynthesis linked to the pathogen's virulence. Its structural data offer a robust framework for identifying small molecules to disrupt sesquiterpene synthesis, attenuating *B. cinerea*'s pathogenicity. The SIX6 protein from *Fusarium oxysporum* f. sp. *lycopersici*(PDB ID: 8EBB), despite its classification as a protein of unknown function, plays a critical role in the virulence mechanism of this vascular wilt pathogen, which devastates crops like tomatoes. Its high-resolution structure at 1.88 Å, with R-free and R-work values of 0.221 and 0.193, supports molecular docking, virtual screening, and lead optimization to design antifungal agents. Finally, the AvrSr35 effector protein from *Puccinia graminis*f. sp. *tritici*(PDB ID: 7XDS), resolved at 2.06 Å with reliable R-values (R-free: 0.287, R-work: 0.257, Observed: 0.258), provides critical insights into wheat stem rust pathogenicity. Its potential as a target for structure-based drug design is underscored by its well-defined active sites and suitability for molecular dynamics simulations and virtual screening. Together, these structural models offer a comprehensive platform for advancing the design of inhibitors to mitigate plant pathogen-induced diseases (Table 1).

*Botrytis cinerea*, a necrotrophic fungus [29], induces gray mold on more than 200 plant species, such as grapes, strawberries, and tomatoes [30], particularly thriving in environments characterized by high humidity and cool temperatures [31]. Its dissemination occurs through airborne conidia and entry into plants is facilitated through either wounds or natural openings [32–35]. The impact on yield can be substantial, with potential losses of up to 50% in crops affected by this pathogen [35–37], leading to significant economic loss estimated in the 10-100 billions of dollars globally on an annual basis [38–40]. *Fusarium oxysporum* f. sp. *lycopersici,* a soil-borne fungal pathogen [41–43], induces Fusarium wilt in tomatoes under favorable moist conditions [41,44]. Its dissemination occurs through contaminated soil, water, and plant matter, gaining access through roots and inhabiting the plant's vascular system [45–47]. Severe infestations can lead to yield reductions in agricultural fields, with notable economic repercussions in tomato cultivation areas globally [48,49]. *Puccinia graminis* f. sp. *tritici*, a fungal pathogen accountable for stem rust in wheat [50], prospers under warm and moist conditions. Its dissemination occurs via airborne urediniospores capable of long-distance travel, infiltrating plants through stomata [51]. The potential yield reductions can be catastrophic, with losses escalating to 100% during severe outbreaks, significantly impacting worldwide wheat production and food security [52]. The rise of highly virulent strains underscoring the necessity for vigorous resistance breeding and comprehensive disease management tactics [53].The ongoing challenges presented by these fungal pathogens and the encouraging outcomes from computational investigations prompt this study to delve deeper into validating and exploring innovative targets for fungicidal activity through an integrated in silico methodology.

**Table 1. Protein information.**

| Title | PDB ID:7VEM | PDB ID:8H6Q | PDB ID:8EBB | PDB ID:7XDS |
|---|---|---|---|---|
| Organism | *Phytophthora capsici* | *Botrytis cinerei* | *Fusarium oxysporum*f. sp.*lycopersici* | *Puccinia graminis*f. sp.*tritici* |
| Resolution | 2.39 Å | 2.00 Å | 1.88 Å | 2.06 Å |
| R-Value Free | 0.224 | 0.195 | 0.221 | 0.287 |
| Ramachandran plot, % | 89.5 | 93.8 | 91.8 | 94.7 |
| References | [57] | [58] | [59] | [60] |

Through the utilization of DFT, molecular docking, and MD simulations, the goal is to pinpoint and enhance potent inhibitors characterized by high specificity and minimal off-target effects, which could potentially facilitate the advancement of next-generation fungicides. This holistic approach holds the potential to enhance our comprehension of fungal pathogen mechanisms and make a significant contribution to the realms of sustainable agriculture and global food security.

## Methods

### Optimization and preparation of ligand

Material Studio 8.0 was used for extensive computational modeling and molecular optimization. The DMol3 algorithm was used for a thorough examination that included the identification of chemical descriptors, quantum characteristics, and geometric optimization. To provide precise and trustworthy findings, this procedure used the DFT functional in conjunction with the BLY3 basis set [54–56]. Utilizing the DFT functional, various quantum properties, includinglowest unoccupied molecular orbital(LUMO), highest occupied molecular orbital (HOMO), energy gap, ionization potential (I), electron affinity (A), chemical potential (μ), electronegativity (χ), hardness (η), softness (s), and electrophilicity (ω) were calculated using equations [5]:

$$E_{gap} = \left( E_{LUMO} - E_{HOMO} \right) \tag{1}$$

$$I = -E_{HOMO} \tag{2}$$

$$A = -E_{LUMO} \tag{3}$$

$$(X) = \frac{I + A}{2} \tag{4}$$

$$(\omega) = \frac{\mu^2}{2\eta} \tag{5}$$

$$(\mu) = -\frac{I + A}{2} \tag{6}$$

$$(\eta) = \frac{I - A}{2} \tag{7}$$

$$(S) = \frac{1}{\eta} \tag{8}$$

### Protein preparation and collection

The meticulous process of selecting and preparing the four fungal proteins, namely 7VEM (*Phytophthora capsica*), 8H6Q (*Botrytis cinerea*), 8EBB (*Fusarium oxysporum* f. sp. *lycopersici.*), and 7XDS (*Puccinia graminis* f. sp. *tritici*), was undertaken to ensure the reliability of the subsequent docking simulations. All of these proteins were found in different plant disease-causing fungi evaluated through the X-ray diffraction method with high stable configuration Ramachandran outliers listed (Table 1). These proteins were obtained from the protein data bank (PDB), which can be accessed at http://www.rcsb.org, on 5th August 2023. The

selection of these proteins was based on their biological relevance, specifically their potential association with the study objectives. In order to create a clean protein template for docking using Discovery Studio, co-crystallized ligands, water molecules, and ions were removed. Additionally, the protein structures underwent energy minimization to optimize their conformations and alleviate any steric clashes. This ensured that the selected fungal proteins were properly prepared for accurate and meaningful molecular docking investigations.

## Molecular docking and visualization of docking

**Molecular docking by PyRx.** The molecular docking analysis was conducted using PyRx software, employing the AutoDock Wizard option. To evaluate the binding affinity between the ligand and individual macromolecules, essential parameters (as detailed in Table 2) were obtained and utilized. In this process, the protein was loaded as a macromolecule, and the ligand was loaded separately. Subsequently, the loaded ligand underwent optimization for maximum energy, considering parameters such as grid surface area, center of grid, and grid dimensions. The goal was to ensure adequate coverage of the total surface area of both the ligand and the protein.The docking process was initiated with the specified parameters outlined (Table 2) within the PyRx software's AutoDock Wizard option. Following the docking procedure, the resulting ligand-protein docking complex was further analyzed for non-covalent interactions using Discovery Studio visualization [61].

**Molecular docking by Glide from Schrodinger suite.** In this research, glide docking was utilized as the molecular docking technique to examine the binding interactions between a variety of ligands and a target protein.The molecular docking was done using Glide tool on Schrodinger suite.For each protein structure, a grid box of $30 \times 30 \times 30$ Å$^3$ with a default inner box ($20 \times 20 \times 20$ Å$^3$) was centered on the corresponding ligand [62]. There were however no boundaries and the setting parameters were set to default. Following that, conformational sampling was put into effect to the ligands, and their anticipated binding affinities were then utilized to evaluate them. The Schrödinger software suite includes the Glide docking tool, which enables a systematic search of the conformational space to facilitate a comprehensive examination of ligand-protein interactions. Considerations like attachment modes, bonding by hydrogen patterns, and energetics were thoughtfully taken into account when analyzing the docking results. This method produced insightful information about the connections between

**Table 2. Grid box parameters used for docking analysis in this study.**

| Protein Name with PDB ID | Grid Box Size | |
| --- | --- | --- |
| | **Center** | **Dimension(Å)** |
| *Phytophthora capsici*(PDB: 7VEM) | X = −0.28 | X = 83.52 |
| | Y = −25.03 | Y = 79.50 |
| | Z = −7.46 | Z = 72.44 |
| *Botrytis cinerea* (PDB: 8H6Q) | X = 47.34 | X = 95.68 |
| | Y = 16.24 | Y = 97.38 |
| | Z = 19.13 | Z = 80.33 |
| *Fusarium oxysporum*f.sp. lycopersici(PDB: 8EBB) | X = 2.77 | X = 70.97 |
| | Y = 17.83 | Y = 56.62 |
| | Z = −11.09 | Z = 67.38 |
| *Puccinia graminis*f. sp. tritici(PDB: 7XDS) | X = 31.47 | X = 63.80 |
| | T = 75.30 | Y = 78.47 |
| | Z = 47.42 | Z = 107.45 |

structure and activity as well as possible binding locations that are essential for the logical design of novel ligands with enhanced pharmacological profiles. By defining the binding (active) site residues, which were found, the binding site receptor grid for plant pathogenic fungal proteins was created. The docked conformers were evaluated using Glide (G) Score. The G Score is calculated as follows in equation [9]:

$$G\ Score = a \times vdW + b \times coul + Lipo + HBond + Metal + BuryP + RotB + Site \quad (9)$$

Wherein, vdW denotes van der Waals energy, Coul denotes Coulomb energy, Lipo denotes lipophilic contact, HBond indicates hydrogen-bonding, Metal indicates metal-binding, BuryP indicates penalty for buried polar groups, RotB indicates penalty for freezing rotatable bonds, Site denotes polar interactions in the active site and the $a = 0.065$ and $b = 0.130$ are coefficients of vdW and Coul.

## Pharmacokinetics and ADMET studies

The absorption, distribution, metabolism, excretion and toxicity is expressed in shortageby ADMET, which are valuable and necessary factors in the fungicide development process [63,64]. The ADMET criterion was obtained by use of the SwissADME and pkCSMonline-tool:http://biosig.unimelb.edu.au/pkcsm/prediction_single/adme_1643650057.59(accessed from the 10thOctober 2023) [65].

## Lipinski rule and pharmacokinetics

SwissADME, accessed on October 9, 2023, was utilized to predict pharmacokinetics and assess fungicide-likeness metrics. This online database, available at http://www.swissadme.ch/index.php [66], is respected for its important and adaptable functions that make information easier to access. In order to fully clarify the fungicide-like properties of different ligands, a large number of pharmacokinetic parameters were determined. These factors included the ligands' lipophilicity, number of bond rotations (NRB), and molecular weight. The investigation also took into account the quantity of hydrogen bond acceptors (HBA) and donors (HBD). A thorough grasp of the ligands' potential as fungicides was attained by closely analyzing these variables.

## Molecular dynamic

Simulations were conducted using the Desmond software suite (Schrödinger Release 2024-1: Desmond Molecular Dynamics System, D. E. Shaw Research, New York, NY, 2024) in accordance with established MD protocols. The molecular system, which consisted of a biologically relevant complex, was prepared using the OPLS-AA force field, and the solvent environment was described using an appropriate water model. To ensure solvent relaxation, the system underwent a staged equilibration process that involved restrained dynamics on the solute. Following this, production MD runs were carried out under NPT conditions, with a finite time step and temperature control. Desmond outputs trajectory files, energy logs, and other data that can be further analyzed or visualized using external tools. Trajectory analysis, performed using Maestro and supplemented by external tools, included measurements, such as root mean square deviation (RMSD), root mean square fluctuation (RMSF), radius of gyration, and energy profiles. Visualization tools were used to examine conformational changes and intermolecular interactions throughout the simulation [67]. This comprehensive computational investigation followed established best practices and parameters, providing a thorough exploration of the dynamic behavior of the molecular system under investigation.

## Results and discussion

### Optimized structure

The study of a computational procedure to determine the quantum calculations of any chemical species requires the optimization of the molecular structure, which is an important aspect of its structural geometry [68]. Additionally, accurate computational parameters are obtained by determining the most stable configuration of any chemical structure. In this study, all compounds underwent computational optimization using the DFT functional, and their primary and most stable configuration was observed with minimal energy required for optimization. The antifungal ligands are Azoxystrobin(L01), Cyproconazole(L02), Difeno-conazole (L03), Tebuconazole (L04), Tricyclazole (L05), Chlorothalonil (L06), Benalaxyl (L07),Bismerthiazol (L08), Carbendazim (L09), Hexaconazole (L10), Thiram (L11), Car-boxin (L12), Iprodione (L13), Kresoxim-Methyl (L14), Cymoxanil (L15), Dichloran (L16), Propiconazole (L17), Dimethomorph (L18), Pyraclostrobin (L19) andAmetoctradin (L20) are shown (Figs 1 and S1).

### HOMO, LUMO, and chemical reactivity descriptors

The calculations of the molecular parameters, including LUMO, HOMO, energy gap ($\Delta E$ gap), chemical potential ($\mu$), electronegativity ($\chi$), hardness ($\eta$), softness ($\sigma$), and electro-philicity ($\omega$), were conducted using the DFT functional, and the results are presented in Table 3. The HOMO-LUMO gap serves as a crucial indicator of chemical reactivity and stability in molecules. In this context, a wider gap suggests higher chemical stability, while a narrower gap implies increased reactivity [69–72]. The findings reveal that the HOMO–LUMO gaps range from 2.474 eV to 10.040 eV across all studied ligands. Notably, ligands L01 exhibit minor energy gaps and minimal softness values, indicating their potential for reactivity and reduced stability. In contrast, L14 stands out with the greatest hardness and the largest energy gap, emphasizing its stability.It is observed that the order of the energy gap is L01 < L10 < L13 < L15 < L18 < L19 < L07 < L02 < L09 < L20 < L03 < L05 < L06 <

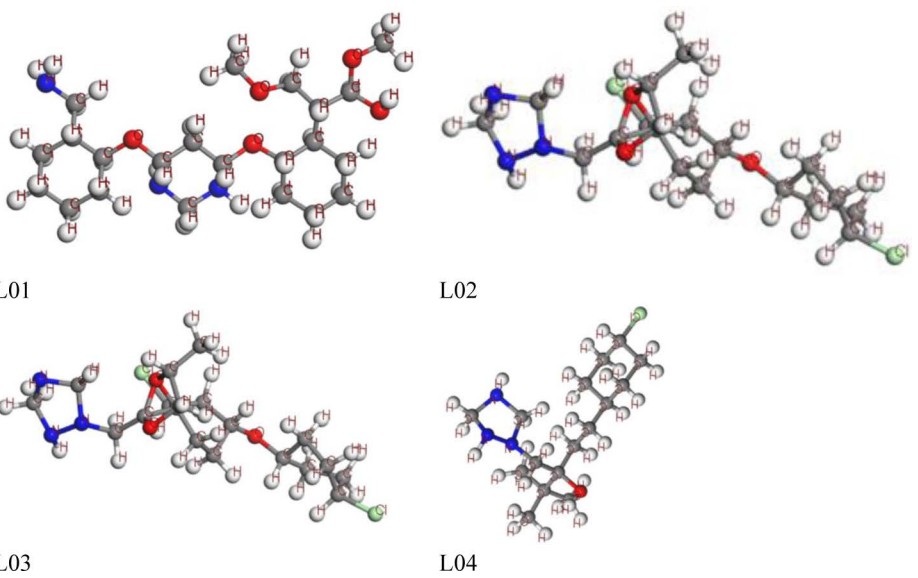

L01

L02

L03

L04

**Fig 1. Optimized structure of inhibitors.**

**Table 3. Data of chemical descriptors.**

| Ligand | LUMO | HOMO | A = −LUMO | I = − HOMO | Energy gap = I−A | Chemical Potential ($\mu$) = −I + A/2 | Hardness ($\eta$) = I−A/2 | Electronegativity ($x$) = I + A/2 | Softness ($\sigma$) = 1/n | Electrophilicity ($\omega$) = $\mu$2/2$\eta$ |
|---|---|---|---|---|---|---|---|---|---|---|
| L01 | 2.131 | −9.534 | −2.131 | 9.534 | 11.665 | −3.701 | 5.832 | 3.701 | 0.171 | 1.174 |
| L02 | 1.517 | −9.034 | −1.517 | 9.034 | 10.551 | −3.758 | 5.275 | 3.758 | 0.189 | 1.338 |
| L03 | 0.829 | −8.111 | −0.829 | 8.111 | 8.940 | −3.641 | 4.470 | 3.641 | 0.223 | 1.482 |
| L04 | 1.361 | −9.076 | −1.361 | 9.076 | 10.437 | −3.857 | 5.218 | 3.857 | 0.191 | 1.425 |
| L05 | 0.043 | −8.943 | −0.043 | 8.943 | 8.986 | −4.450 | 4.493 | 4.450 | 0.222 | 2.203 |
| L06 | 0.661 | −9.386 | −0.661 | 9.386 | 10.047 | −4.362 | 5.023 | 4.362 | 0.199 | 1.894 |
| L07 | 1.893 | −8.727 | −1.893 | 8.727 | 10.620 | −3.417 | 5.310 | 3.417 | 0.188 | 1.099 |
| L08 | −1.074 | −9.382 | 1.074 | 9.382 | 8.308 | −5.228 | 4.154 | 5.228 | 0.240 | 3.289 |
| L09 | 1.661 | −9.441 | −1.661 | 9.441 | 11.102 | −3.890 | 5.551 | 3.890 | 0.180 | 1.363 |
| L10 | 0.819 | −8.843 | −0.819 | 8.843 | 9.662 | −4.012 | 4.831 | 4.012 | 0.207 | 1.665 |
| L11 | −2.417 | −9.328 | 2.417 | 9.328 | 6.911 | −5.872 | 3.455 | 5.872 | 0.289 | 4.990 |
| L12 | 0.201 | −8.810 | −0.201 | 8.810 | 9.011 | −4.304 | 4.505 | 4.304 | 0.221 | 2.056 |
| L13 | 1.176 | −9.643 | −1.176 | 9.643 | 10.819 | −4.233 | 5.409 | 4.233 | 0.184 | 1.656 |
| L14 | −1.003 | −7.401 | 1.003 | 7.401 | 6.398 | −4.202 | 3.199 | 4.202 | 0.312 | 2.759 |
| L15 | 1.375 | −9.528 | −1.375 | 9.528 | 10.903 | −4.076 | 5.451 | 4.076 | 0.183 | 1.524 |
| L16 | −2.474 | −10.040 | 2.474 | 10.040 | 7.566 | −6.257 | 3.783 | 6.257 | 0.264 | 5.174 |
| L17 | 1.240 | −8.880 | −1.240 | 8.880 | 10.120 | −3.820 | 5.060 | 3.820 | 0.197 | 1.441 |
| L18 | 1.451 | −9.976 | −1.451 | 9.976 | 11.427 | −4.262 | 5.713 | 4.262 | 0.175 | 1.590 |
| L19 | 1.241 | −9.036 | −1.241 | 9.036 | 10.277 | −3.897 | 5.138 | 3.897 | 0.194 | 1.478 |
| L20 | 1.578 | −9.137 | −1.578 | 9.137 | 10.715 | −3.779 | 5.357 | 3.779 | 0.186 | 1.333 |

L17 < L08 < L12 < L04 < L14 < L11 < L16. (Table 3) further illustrates that the softness values are approximately 0.228 or less than 0.30, underscoring the potential for faster degradation and disintegration for elements with higher softness values [73–76]. Conversely, hardness, a crucial stability indicator, is reflected in the compounds' resistance to changes in electron configuration [77–79]. Higher hardness values signify increased stability and resistance to changes, providing valuable insights into the chemical behavior of these compounds.

## Frontier Molecular Orbital: HOMO and LUMO

The frontier molecular orbital (FMO) was used to assess the kinetics, and the engaged regions where the protein could be folded become the active pharmacophore or active functional group. The maximum energy chemical orbital that an electron can occupy in a particular molecule is called the HOMO. Various chemical reactions include electrons in the HOMO, especially electron transfer processes. In addition to acting as the donor of electrons in reactions, the HOMO plays a critical role in nucleophilic assaults by contributing its electron density [80–84]. The lowest energy molecular orbital in a molecular structure that is free of electrons is known as the LUMO. During reactions, the LUMO and electrons from the HOMO of another molecule regularly interact. In electrophilic assaults, where it accepts electrons, the LUMO plays a crucial role as the electron acceptor during reactions. The dark blue color indicates the positive terminal of the orbitals in both LUMO, HOMO, while the pink color denotes the negative node. The more minor energy gap assists in the development of fungicide interaction with a protein. From the pictures in (Figs 2 and S2), there is various part of different molecules for HOMO and LUMO.

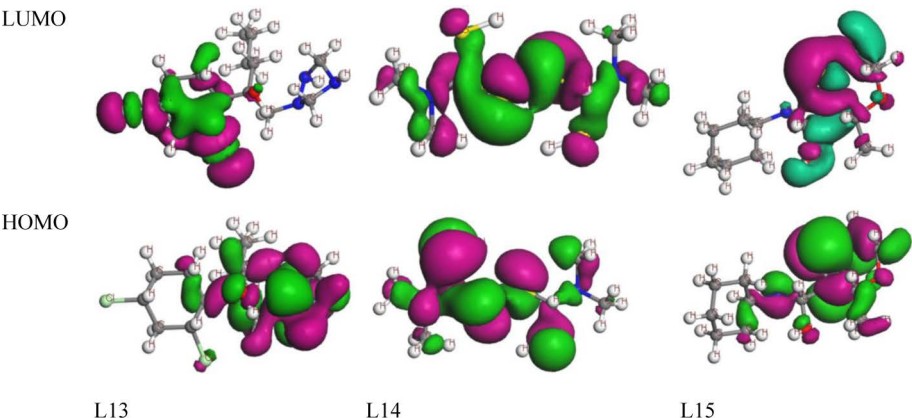

**Fig 2. Frontier molecular orbitals diagram for HOMO and LUMO.**

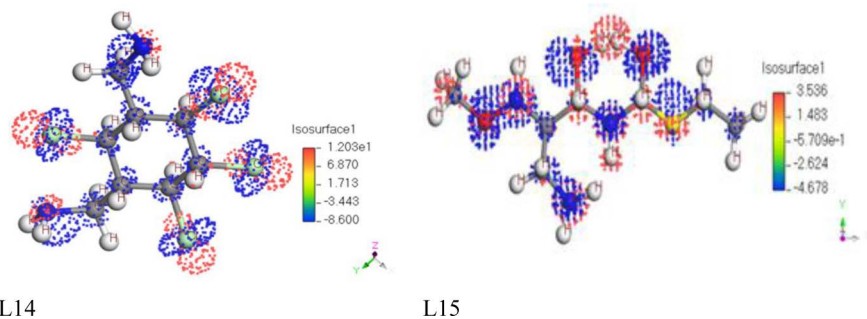

**Fig 3. Electrostatic potential map for Frontier Molecular Orbital of L14 and L15.**

## Electrostatic potential map for Frontier Molecular Orbital

Frontier Molecular Orbitals (FMOs) are a set of molecular orbitals that play a crucial role in the reactivity of a molecule. The two most important FMOs are the HOMO and the LUMO showed in the (Figs 3 and S3). These orbitals are involved in various chemical reactions and properties, especially in the context of organic chemistry. Electrostatic potential maps for the HOMO and LUMO can provide valuable insights into the reactivity and electronic structure of a molecule.

## Molecular docking

**Auto docking by PyRx.** Molecular docking simulations were conducted to authenticate the pharmacological findings. These simulations confirmed the binding of fungicide compounds with the associated peptide in the presence of four-plant pathogenic fungal proteins, namely *Phytophthora capsica*, *Botrytis cinerea*, *Fusarium oxysporum* f. sp. *lycopersici*, and *Puccinia graminis* f. sp. *tritici*. The primary cause of binding between the compounds and proteins is the hydrogen and hydrophobic bonds, which are demonstrated by molecular docking. The interaction between protein and ligand is crucial for the development of structurally oriented fungicides. It is widely accepted that docking scores higher than -6.00 kcal/mol indicate a standard fungicide [85,86].Additionally, molecular docking is a reliable approach for understanding the engagement of two molecules and identifying the optimal configuration for

ligand binding. Through *insilico* experiments, it was revealed that the fungicide compounds in Table 4 exhibit excellent binding affinity to the target proteins, with values ranging from −3.00 to −9.20 kcal/mol. Among the four fungal pathogens, the highest docking score of −8.20 kcal/mol was observed for *Phytophthora capsici*(7vem) in L03, while the lowest score of −3.80 kcal/mol was observed in L11. For *B. cinerei*(8H6Q), the highest docking score of −8.90 kcal/mol was found in L03, and the lowest score of −4.00 kcal/mol was found in L11. For *Fusarium oxysporum* f. sp. *lycopersici*(8EBB), the highest docking score of −8.40 kcal/mol was found in L03, and the lowest score of -3.80 kcal/mol was found in L11. Lastly, for *Puccinia graminis* f. sp. *tritici*(7XDS), the highest docking score of −9.20 kcal/mol was found in L18, and the lowest score of −4.30 kcal/mol was found in L11. These findings indicate that the fungicides (L01, L03, L04, L13, L14, L17, L18, and L19) exhibit significantly higher binding affinities against the four fungal pathogenic proteins compared to other ligands.

## Molecular docking by Glide from Schrodinger suite

Schrodinger suite ligand preparation product, Ligprep was used to prepare high quality, all atom 2D structures. The ligand preparation included 2D–3D conversions, generating variations, correction, verification and optimization of the structures. Receptor grid was generated using Receptor grid generation in the Glide application of Maestro (Schrödinger Release 2024-1: Glide, Schrödinger, LLC, New York, NY, 2024). Through in silico experiments, it was revealed that the fungicide compounds in Table 5 exhibit excellent binding affinity to the target proteins, with values ranging from −2.692 to −7.317 kcal/mol. Among the four fungal pathogens, the highest docking score of −7.317 kcal/mol was observed for *Phytophthora capsica* (7vem) in L03, while the lowest score of −3.184 kcal/mol was observed in L20. For *B. cinerei*(8H6Q), the highest docking score of −7.270 kcal/mol was found inL09, and the lowest

**Table 4. Data of binding energy and name of interacted ligand against four fungal proteins in binding affinity (kcal/mol).**

| S.L | Compound | *Phytophthora capsica* (7VEM) | *B. cinerei*(8H6Q) | *Fusarium oxysporum*f. sp. *lycopersici* (8EBB) | *Puccinia graminis f.* sp. *Tritici*(7XDS) |
|---|---|---|---|---|---|
| L01 | Azoxystrobin | −7.10 | −8.20 | −8.10 | −7.60 |
| L02 | Cyproconazole | −6.80 | −7.70 | −6.60 | −7.90 |
| L03 | Difenoconazole | −8.20 | −8.90 | −8.40 | −7.80 |
| L04 | Tebuconazole | −6.80 | −7.80 | −7.20 | −6.70 |
| L05 | Tricyclazole | −6.50 | −7.40 | −6.30 | −7.00 |
| L06 | Chlorothalonil | −6.00 | −6.20 | −5.90 | −5.90 |
| L07 | Benalaxyl | −6.50 | −8.50 | −6.90 | −6.70 |
| L08 | Bismerthiazol | −6.10 | −6.20 | −6.20 | −6.20 |
| L09 | Carbendazim | −6.30 | −7.30 | −6.40 | −7.00 |
| L10 | Hexaconazole | −6.30 | −7.40 | −6.40 | −6.60 |
| L11 | Thiram | −3.80 | −4.00 | −3.80 | −4.30 |
| L12 | Carboxin | −6.60 | −7.20 | −6.10 | −7.20 |
| L13 | Iprodione | −8.10 | −8.10 | −6.80 | −7.90 |
| L14 | Kresoxim-Methyl | −6.60 | −8.50 | −7.00 | −8.30 |
| L15 | Cymoxanil | −5.60 | −6.10 | −5.60 | −6.10 |
| L16 | Dichloran | −5.30 | −5.90 | −5.50 | −5.90 |
| L17 | Propiconazole | −7.20 | −7.90 | −7.20 | −8.40 |
| L18 | Dimethomorph | −7.60 | −8.20 | −7.60 | −9.20 |
| L19 | Pyraclostrobin | −8.00 | −8.40 | −8.10 | −8.70 |
| L20 | Ametoctradin | −6.20 | −7.30 | −5.50 | −6.50 |

**Table 5. Data of binding energy and name of interacted ligand against 4 plant fungal pathogens in binding affinity (kcal/mol).**

| S.L | Compound | *Phytophthora capsici*(7vem) | *Botrytis cinerea*(8H6Q) | *Fusarium oxysporum*f.sp. *Lycopersici*(8EBB) | *Puccinia graminis*f.sp. *tritici*(7XDS) |
|-----|----------|------------------------------|--------------------------|-----------------------------------------------|------------------------------------------|
| L01 | Azoxystrobin | −7.044 | −4.631 | −4.631 | −5.392 |
| L02 | Cyproconazole | −6.074 | −6.109 | −5.487 | −5.317 |
| L03 | Difenoconazole | −7.317 | −6.499 | −5.779 | −5.335 |
| L04 | Tebuconazole | −7.061 | −6.731 | −5.184 | −5.737 |
| L05 | Tricyclazole | −5.633 | −5.707 | −5.648 | −4.332 |
| L06 | Chlorothalonil | −5.137 | −5.603 | −5.065 | −3.899 |
| L07 | Benalaxyl | −5.457 | −4.494 | −6.418 | −2.692 |
| L08 | Bismerthiazol | −4.294 | −3.781 | −3.786 | −3.729 |
| L09 | Carbendazim | −5.582 | −7.270 | −6.042 | −4.442 |
| L10 | Hexaconazole | −5.555 | −6.498 | −5.712 | −3.456 |
| L11 | Thiram | −4.276 | −5.294 | −4.099 | −3.065 |
| L12 | Carboxin | −5.760 | −6.024 | −4.946 | −2.933 |
| L13 | Iprodione | −5.580 | −5.331 | −5.490 | −3.893 |
| L14 | Kresoxim-Methyl | −5.132 | −4.403 | −4.896 | −2.448 |
| L15 | Cymoxanil | −3.976 | −3.830 | −3.317 | −3.501 |
| L16 | Dichloran | −5.585 | −5.675 | −5.370 | −4.097 |
| L17 | Propiconazole | −6.068 | −5.624 | −5.468 | −3.701 |
| L18 | Dimethomorph | −6.068 | −5.624 | −5.468 | −5.285 |
| L19 | Pyraclostrobin | −6.310 | −5.697 | −5.719 | −3.701 |
| L20 | Ametoctradin | −3.184 | −4.232 | −4.123 | −3.828 |

score of −3.317 kcal/mol was found in L15.For *Fusarium oxysporum* f. sp. *lycopersici* (8EBB), the highest docking score of −6.418 kcal/mol was found in L07, and the lowest score of −3.800 kcal/mol was found in L11. Lastly, for *Puccinia graminis* f. sp. *tritici*(7XDS), the highest docking score of −5.737 kcal/mol was found in L04, and the lowest score of −2.448 kcal/mol was found in L14.

## A comparative study for docking results

The comparative analysis of docking results obtained from AutoDock by PyRx and Molecular Docking by Glide from the Schrödinger suite underscores the robust binding affinities of certain fungicide compounds with plant pathogenic fungal proteins. In (Fig 4), AutoDock yielded docking scores ranging from −3.00 to −9.20 kcal/mol, with the highest affinity observed for *Phytophthora capsici*(7vem) with Ligand L03 at −8.20 kcal/mol. Glide results corroborated this finding, producing scores from −2.692 to −7.317 kcal/mol and identifying ligand L03 as having the highest affinity for *Phytophthora capsica* with a score of −7.317 kcal/mol. Notably, both methods consistently identified L03 as having the strongest binding affinity for this pathogen, highlighting its potential efficacy.

Considering the biological systems in contact with these molecules, ligand L03's high affinity suggests it could effectively inhibit the growth and proliferation of *Phytophthora capsici*, a significant plant pathogen. The interaction of L03 with the fungal proteins disrupt critical biological processes within the fungus, thereby mitigating infection and disease spread in plants. However, it is crucial to evaluate the potential impact of L03 on non-target organisms, including beneficial soil microbes, other plant species, and possible environmental persistence. The favorable binding scores indicate that L03 can be developed as an effective fungicide, but comprehensive studies on its environmental behavior, toxicity to non-target species, and overall safety are essential to ensure sustainable and safe agricultural practices. This holistic approach

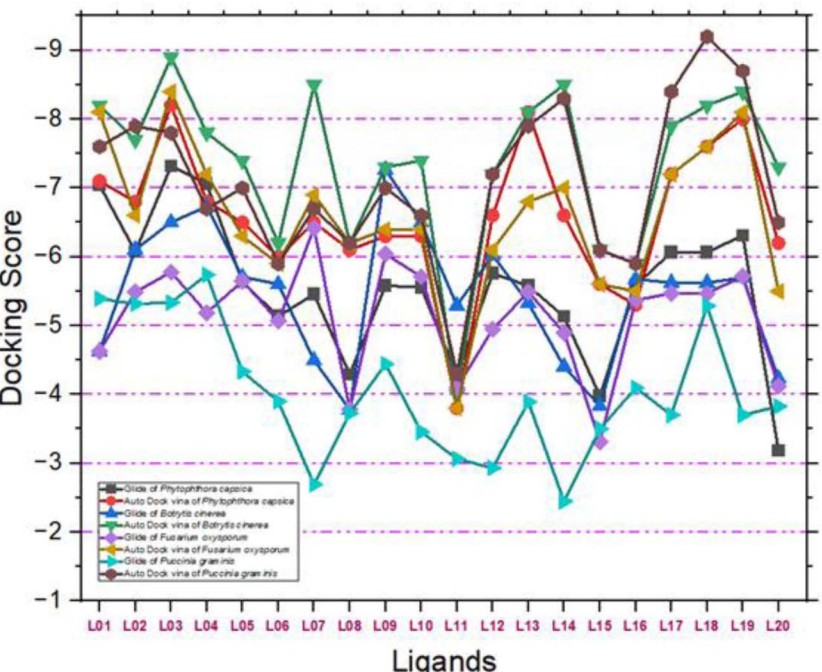

**Fig 4. A comparative study of AutoDock by PyRx and Molecular Docking by Glide by Schrödinger suite.**

will help in leveraging L03's potential while minimizing adverse ecological impacts, thereby contributing to effective integrated pest management strategies.

## Protein-Ligand interaction for auto docking

The most important factor to take into account when determining which fungicide is most effective is the ligand-protein interaction, which is achieved through the formation of weak bonds or covalent bonds. This interaction provides an approximation of the binding affinity or energy of substances with the proteins of micro pathogens where the molecular docking poses of four plant fungal pathogen proteins with 20 anti-fungal ligands mentioned in (Fig 5). The bond distance was assessed to have a better understanding of the interaction between the molecule and the protein associated with the chosen fungal infections. Substitute data indicates that there are various types of bonds, including hydrophilic, hydrophobic, Van der Waal, and H-bonds. Furthermore, the protein's locations that the ligand binds are identified. According to the results, the ligand L03 has the most binding sides, with an H-bond count of five and six hydrophobic bonds against the *B. cinerei*- 8H6Q protein.

## Different poses of docking for protein ligand interaction

**PyRx Docking.** The ligand binding sites with receptor was identified with the help of Discovery Studio version 2020, and graphically represents (Figs 6–9) as well as (S4 Fig). In this case, at first, auto docking has been performed on the protein and ligand to identify the binding sites and obstructing the active site, as well as determining the amino acid residue. The mostly present bond is hydrogen and hydrophobic bond, and they are responsible for docking score variation. After performing molecular docking to predict the interaction between a drug and a protein pocket, several post-docking steps and analyses are typically carried out to understand and evaluate the results. If available, docking results are frequently

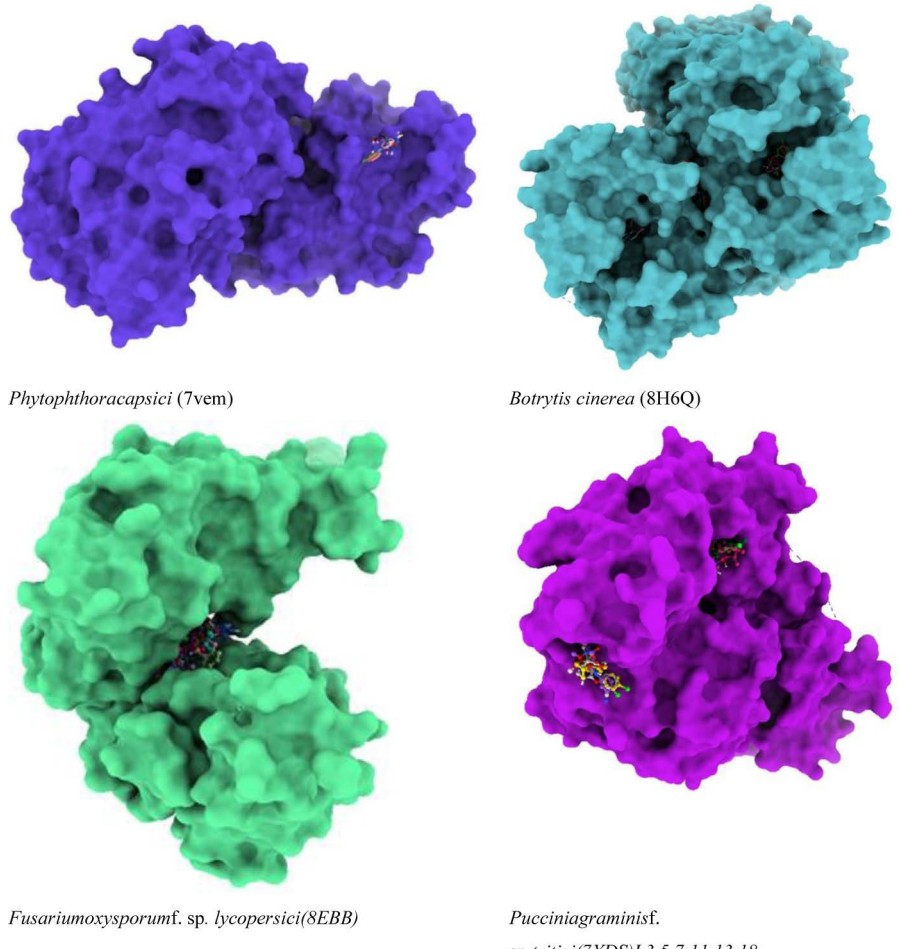

*Phytophthoracapsici* (7vem)

*Botrytis cinerea* (8H6Q)

*Fusariumoxysporum*f. sp. *lycopersici(8EBB)*

*Pucciniagraminis*f. sp.*tritici(7XDS)L3,5,7,11,13,18*

**Fig 5. Molecular docking poses of four plant fungal pathogen proteins with 20 anti-fungal ligands.**

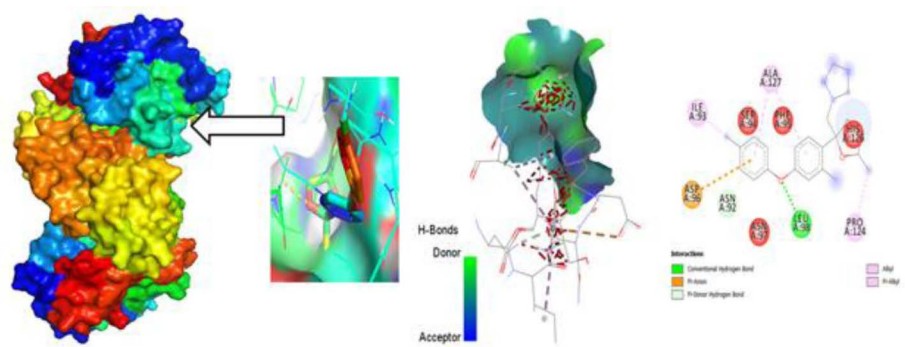

**Fig 6. Molecular docking poses of plant pathogenic fungal protein *Phytophthoracapsici* (PDB: 7vem) with L03.**

compared to experimental data to verify the precision of the predictions. A comparison of anticipated binding modes with crystallographic or other experimental structures may be used for validation. To make sure it predicts known binding affinities appropriately, the scoring

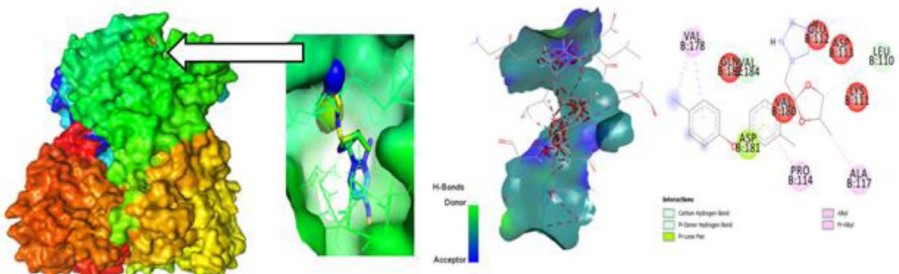

**Fig 7. Molecular docking poses of plant pathogenic fungal protein *Botrytis cinerea (*PDB: 8H6Q) with L03.**

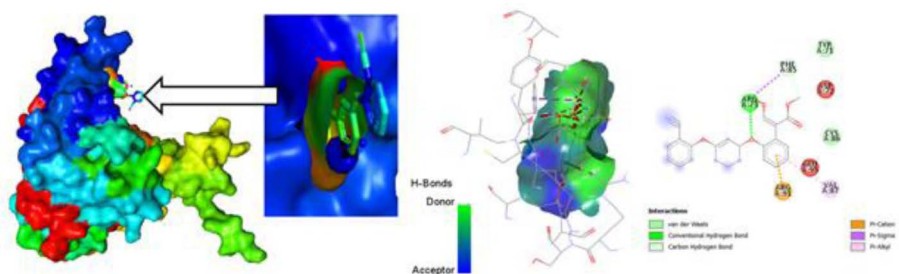

**Fig 8. Molecular docking poses of plant pathogenic fungal protein *Fusarium oxysporum* f. sp. Lycopersici(PDB: 8EBB) withL01.**

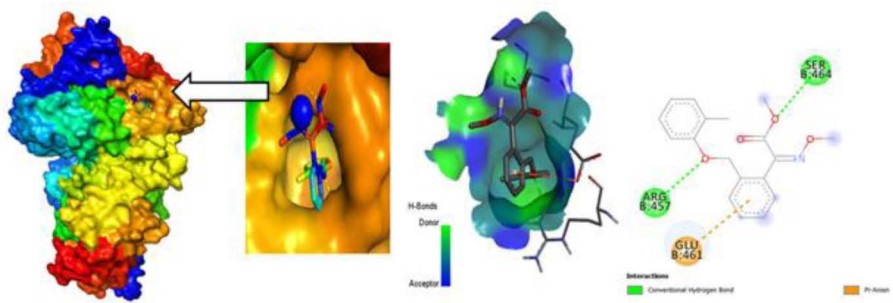

**Fig 9. Molecular docking poses of plant pathogenic fungal protein *Pucciniagraminis* f. sp. *tritici (*PDB: 7XDS) with L18.**

function utilized during docking is assessed. Future projections may be more accurate if relevant improvements are made. So, what, all drugs are in the molecular pocket of proteins, so the docking procedure, several post-docking steps and analyses are valid.

## Protein ligand interaction from Glide docking

Glide, developed by Schrödinger, is a widely used molecular docking program that predicts the binding mode and affinity of a small molecule (ligand) to a target protein. Analyzing protein-ligand interactions from Glide docking involves examining the docking results and visualizing the binding mode exhibited (Fig 10). Consider the docking score provided by Glide as an estimate of the binding affinity. Lower scores generally indicate better binding.

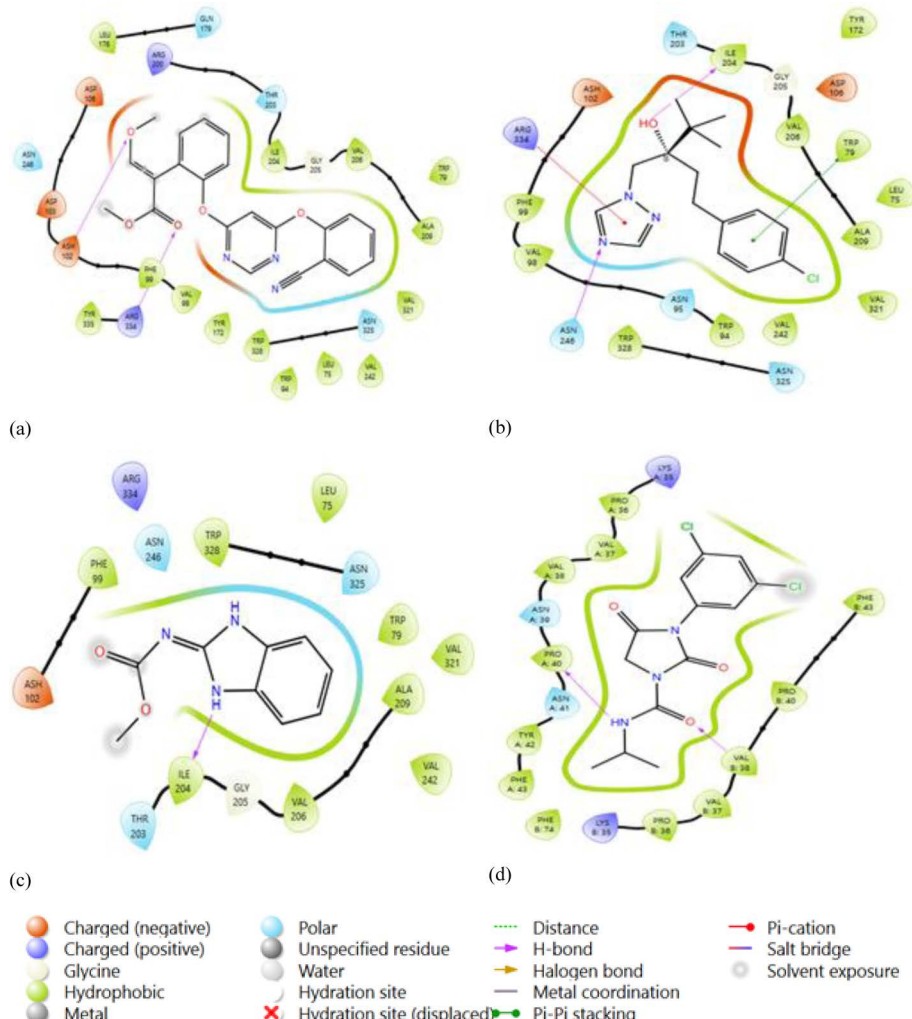

**Fig 10. 2D stick diagram of (A) Azoxystrobin; (B) Tebuconazole; (C) Carbendazim; (D) Iprodione; illustrating hydrogen bonds and pi-pi stacking formed with the amino-acid residues at the binding pocket of Botrytis cinerea (8H6Q).**

After performing molecular docking to predict the interaction between a drug and a protein pocket, several post-docking steps and analyses are typically carried out to understand and evaluate the results. Numerous binding poses, or hypothesized binding configurations, have been generated for the drug by the docking software inside the protein pocket. A score has been allocated to each pose, which symbolizes the interaction's expected binding energy or inclination. On average, lower scores indicate more favorable binding. To make sure it predicts known binding affinities appropriately, the scoring function utilized during docking is assessed. Future projections may be more accurate if relevant improvements are made.

## Pharmacokinetics and fungicide-likeness study

Pharmacological fungicide-likeness is a groundbreaking evaluation of the potential of a particular chemical used as an oral medicine with respect to bioavailability. It is estimate that nine out of twelve targeted medicines are not transparently changed due to their negative effect, resulting in significant medication costs, time, and human resources being wasted [87]. This

problem occurs due to failure to identify the actual fungicide characteristics. However, by employing a new approach, Lipinski's five-rule, it is possible to readily test the aspects of lead compounds, such as their bioavailability and G.I. absorption, among other things [88].

From Table 6 it can be seen that all the ligands follow the Lipinski's five-rule so it can be said that they can remain active as potential drug as small molecules. Each drug can display 0.55 bioavailability score, and they do not violate any conditions of Lipinski's five-rule.

## Bio-availability and toxicity of the fungicides

The bioavailability and toxicity profiles of the twenty ligands reveal significant variations in their absorption, permeability, blood-brain barrier (BBB) penetration, and interactions with P-glycoprotein, cation transporters, and CYP450 enzymes. Most ligands exhibit high absorption in the human intestine, with L02, L03, L04, L05, L09, L12, L13, L14, L15, and L20 approaching or achieving near-complete absorption added in Table 7. Ligand L05 shows excellent BBB penetration, indicating potential central nervous system activity, whereas L01 and L02 exhibit moderate permeability. The interaction with P-glycoprotein varies, with L11 and L12 showing significant inhibition, which could influence drug-drug interactions and efflux. Additionally, the renal organic cation transporter interactions highlight L11 and L15 as noteworthy due to their high scores. Most ligands are substrates for CYP450 enzymes, with L01, L04, L05, L10, L14, L15, L17, and L18 showing notable inhibitory activity, suggesting potential for metabolic interactions. The sub-cellular localization data further suggest diverse intracellular distribution, with L01, L05, L07, L08, L10, L12, and L20 showing significant localization, possibly indicating targeted intracellular effects. Overall, L11 and L12 emerge as particularly potent candidates given their comprehensive favorable profiles across multiple

**Table 6. Data of Lipinski rule, pharmacokinetics, and fungicide likeness.**

| Ligand No | Compound | Molecular weight, g/mol | Number of Rotatable bonds | Hydrogen Bond acceptor | Hydrogen Bond donor | Topological polar surface area (Å²) | Lipinski rule result | viola-tion | Bioavail-ability score |
|---|---|---|---|---|---|---|---|---|---|
| L01 | Azoxystrobin | 403.39 | 8 | 8 | 2 | 103.56 | Yes | 0 | 0.56 |
| L02 | Cyproconazole | 291.78 | 5 | 3 | 0 | 50.94 | Yes | 0 | 0.55 |
| L03 | Difenoconazole | 406.26 | 5 | 5 | 1 | 58.40 | Yes | 0 | 0.55 |
| L04 | Tebuconazole | 307.82 | 6 | 3 | 0 | 50.94 | Yes | 0 | 0.55 |
| L05 | Tricyclazole | 189.24 | 0 | 2 | 1 | 58.43 | Yes | 0 | 0.55 |
| L06 | Chlorothalonil | 265.91 | 0 | 2 | 1 | 47.58 | Yes | 0 | 0.55 |
| L07 | Benalaxyl | 325.40 | 7 | 3 | 0 | 46.61 | Yes | 0 | 0.55 |
| L08 | Bismerthiazol | 278.40 | 4 | 2 | 2 | 202.08 | Yes | 0 | 0.55 |
| L09 | Carbendazim | 191.19 | 3 | 3 | 1 | 67.01 | Yes | 0 | 0.55 |
| L010 | Hexaconazole | 314.21 | 6 | 3 | 0 | 50.94 | Yes | 0 | 0.55 |
| L011 | Thiram | 240.43 | 5 | 0 | 0 | 121.26 | Yes | 0 | 0.55 |
| L012 | Carboxin | 235.30 | 3 | 2 | 0 | 63.63 | Yes | 0 | 0.85 |
| L013 | Iprodione | 330.17 | 4 | 3 | 1 | 69.72 | Yes | 0 | 0.55 |
| L014 | Kresoxim-Methyl | 313.35 | 7 | 5 | 0 | 57.12 | Yes | 0 | 0.55 |
| L015 | Cymoxanil | 198.18 | 6 | 5 | 1 | 103.58 | Yes | 0 | 0.55 |
| L016 | Dichloran | 207.01 | 1 | 2 | 1 | 71.84 | Yes | 0 | 0.55 |
| L017 | Propiconazole | 342.22 | 5 | 4 | 1 | 49.17 | Yes | 0 | 0.55 |
| L018 | Dimethomorph | 387.86 | 6 | 4 | 0 | 48.00 | Yes | 0 | 0.55 |
| L019 | Pyraclostrobin | 387.82 | 8 | 5 | 0 | 65.82 | Yes | 0 | 0.55 |
| L20 | Ametoctradin | 275.39 | 8 | 3 | 0 | 69.10 | Yes | 0 | 0.55 |

**Table 7. Bio-availability and toxicity of the fungicides.**

| Ligand | Absorption Human Intestinal | Caco-2 Permeability | Blood-Brain Barrier (BBB) | P- I glycopro-tein inhibitor | Cation Transporter Renal Organic | P- II glycopro-tein substrate | Sub-cellular localization | Substrate CYP450 2C9 | Inhibitor CYP450 1A2 |
|---|---|---|---|---|---|---|---|---|---|
| L01 | 0.5107 | 0.5663 | 0.6630 | 0.7237 | 0.6170 | 0.5442 | 0.8384 | 0.8069 | 0.7507 |
| L02 | 0.9812 | 0.5617 | 0.7017 | 0.6006 | 0.7102 | 0.8528 | 0.5908 | 0.7075 | 0.7079 |
| L03 | 0.9674 | 0.5825 | 0.6071 | 0.7615 | 0.6265 | 0.8671 | 0.4655 | 0.8009 | 0.6706 |
| L04 | 0.9745 | 0.5742 | 0.5549 | 0.8515 | 0.6636 | 0.8712 | 0.6484 | 0.7393 | 0.7171 |
| L05 | 0.9915 | 0.5075 | 0.9372 | 0.5704 | 0.5913 | 0.6617 | 0.5332 | 0.8212 | 0.6994 |
| L06 | 0.8851 | 0.5476 | 0.9355 | 0.6096 | 0.7651 | 0.7811 | 0.5515 | 0.7726 | 0.5323 |
| L07 | 0.8963 | 0.6421 | 0.9567 | 0.8274 | 0.7913 | 0.7518 | 0.5677 | 0.7434 | 0.7794 |
| L08 | 0.6986 | 0.5799 | 0.9532 | 0.5613 | 0.7639 | 0.7321 | 0.5078 | 0.8535 | 0.8911 |
| L09 | 0.9861 | 0.5288 | 0.9454 | 0.7666 | 0.6042 | 0.6150 | 0.6030 | 0.7474 | 0.5490 |
| L10 | 0.9724 | 0.5886 | 0.5579 | 0.6053 | 0.5000 | 0.9115 | 0.4643 | 0.6683 | 0.6965 |
| L11 | 0.9625 | 0.5000 | 0.9664 | 0.9950 | 0.9064 | 0.8586 | 0.4894 | 0.8228 | 0.5176 |
| L12 | 0.9960 | 0.5708 | 0.9606 | 0.9479 | 0.7373 | 0.6943 | 0.6446 | 0.6602 | 0.5136 |
| L13 | 0.0992 | 0.6004 | 0.6893 | 0.9696 | 0.8378 | 0.7551 | 0.6052 | 0.7278 | 0.6798 |
| L14 | 0.9930 | 0.5480 | 0.9247 | 0.7869 | 0.6053 | 0.6295 | 0.7294 | 0.7948 | 0.8582 |
| L15 | 0.9971 | 0.6077 | 0.9144 | 0.7309 | 0.9168 | 0.6570 | 0.8082 | 0.8151 | 0.739 |
| L16 | 0.5153 | 0.5753 | 0.7250 | 0.7997 | 0.7941 | 0.5927 | 0.4554 | 0.7457 | 0.5622 |
| L17 | 0.7399 | 0.5911 | 0.6208 | 0.9683 | 0.6607 | 0.8207 | 0.4652 | 0.8073 | 0.6451 |
| L18 | 0.9758 | 0.5269 | 0.7663 | 0.8813 | 0.5153 | 0.8390 | 0.5495 | 0.8533 | 0.7283 |
| L19 | 0.9799 | 0.6043 | 0.6700 | 0.6214 | 0.7787 | 0.5211 | 0.6468 | 0.6860 | 0.6461 |
| L20 | 1.0000 | 0.5205 | 0.9108 | 0.6888 | 0.5000 | 0.5886 | 0.3373 | 0.8896 | 0.5000 |

parameters, while others like L01 and L09 demonstrate lower permeability and reactivity, potentially impacting their efficacy and toxicity profiles.

## Molecular dynamics

**Root Mean Square Deviation (RMSD).** For exploring the structural inflexibility and validating the docking scenarios of the topmost ligand-protein complexes *Phytophthora capsici*(7vem) – Difenoconazole, *Botrytis cinerea* (8H6Q) – Carbendazim, *Fusarium oxysporum* f. sp. *Lycopersici* (8EBB) - Difenoconazole, *Puccinia graminis* f. sp. *triti*ci (7XDS), MD simulations were performed for 100 ns. By analyzing the C-alpha atom's RMSD, the firmness of the ligand-protein complexes was assessed.

Protein complexes exhibit initial fluctuations, followed by a transition to a stable state. When the RMSD values fall below 1.8 Å (Angstroms), it typically suggests a robust structural alignment or similarity between the compared structures (Fig 11). An RMSD below 1.8 Å signifies a significant degree of structural similarity, indicating that the examined structures closely resemble each other in terms of both overall form and atomic positions. Attaining RMSD values below 1.8 Å is commonly regarded as an indication of the precision of predicted structures in simulation and molecular modeling studies. This implies a close resemblance between the modeled structure and the reference or experimental structure.

***Root Mean Square Fluctuation (RMSF) with respect of residues.*** The RMSF with respect to residues is a measure commonly used in the analysis of MD simulations or experimental data of proteins or other biomolecules. It provides insights into the flexibility or mobility of individual amino acid residues within a protein structure. RMSF is calculated as the square root of the average of the squared displacements (or fluctuations) of each residue from its

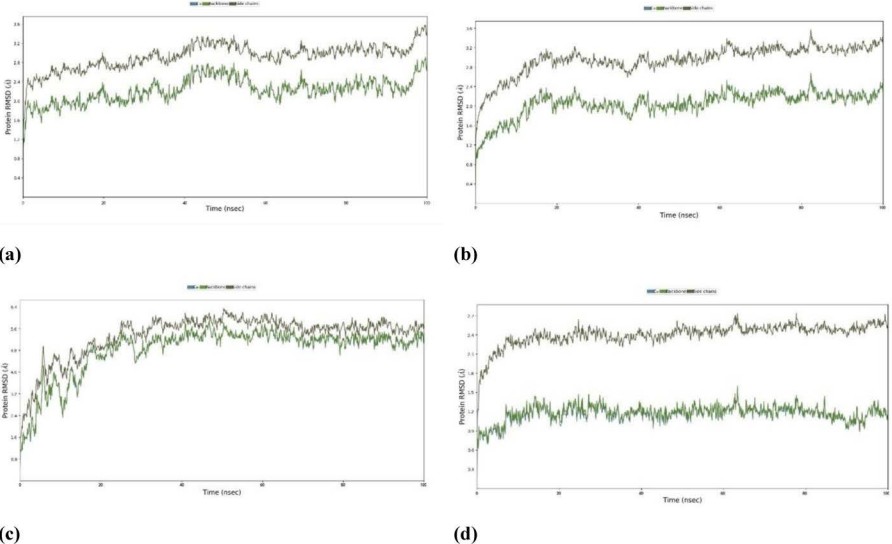

**Fig 11. Root Mean Square Deviation (RMSD) of protein-ligand complexes. complexes** (a) Phytophthoracapsici (7vem) – Difenoconazole, (b) Botrytis cinerea (8H6Q) – Carbendazim, (c) Fusariumoxysporum f. sp. Lycopersici (8EBB) –Difenoconazole, (d) Pucciniagraminis f. sp. tritici (7XDS).

average position over the course of a simulation or experiment. Mathematically, for a protein with N residues, the RMSF of residue i (RMSF and RMSFi) is calculated as:

$$RMSFi = \sqrt{\frac{1}{n}\sum\nolimits_{j=1}^{n}(x_{ij} - \overline{x}_i)^2}$$

Where, $n$ is the number of frames or snapshots in the simulation or experimental data, $xij$ is the position of residue, $i$ in frame j, and $x^-i$ is the average position of residue i over all frames.

In addition, the high RMSF values for certain residues indicate that those residues are more flexible or dynamic. Low RMSF values suggest that those residues are more rigid or less dynamic. RMSF values can be correlated with structural features, functional regions, or interactions within the protein. For example, loops and flexible regions tend to have higher RMSF values compared to core secondary structure elements such as alpha helices and beta strands. It is seen (Fig 12) that the RMSF shows the much less value mean that they are not only excess the value of 2.5 or more. So that it is said that the docked configuration more stable and may be acted as drug after docking accounting by docking. However, it is a valid procedure.

**Protein secondary structure elements (SSE).** Protein secondary structure elements (SSEs) are recurring patterns in protein structures that arise from hydrogen bonding interactions between amino acid residues shown (Fig 13). Common SSEs include alpha helices, beta strands, and loops or turns. In MD analysis, identifying and analyzing SSEs can provide insights into the dynamic behavior and stability of proteins. SSEs can be identified using various algorithms or methods. Commonly used algorithms include DSSP (Define Secondary Structure of Proteins), STRIDE (Structural Identification), and VMD (Visual Molecular Dynamics) among others. These algorithms assign secondary structure elements to each residue in a protein based on the local hydrogen bonding patterns and dihedral angles. Alpha helices are characterized by a regular pattern of hydrogen bonds between residues, resulting in a helical structure. Beta strands form when adjacent segments of the polypeptide

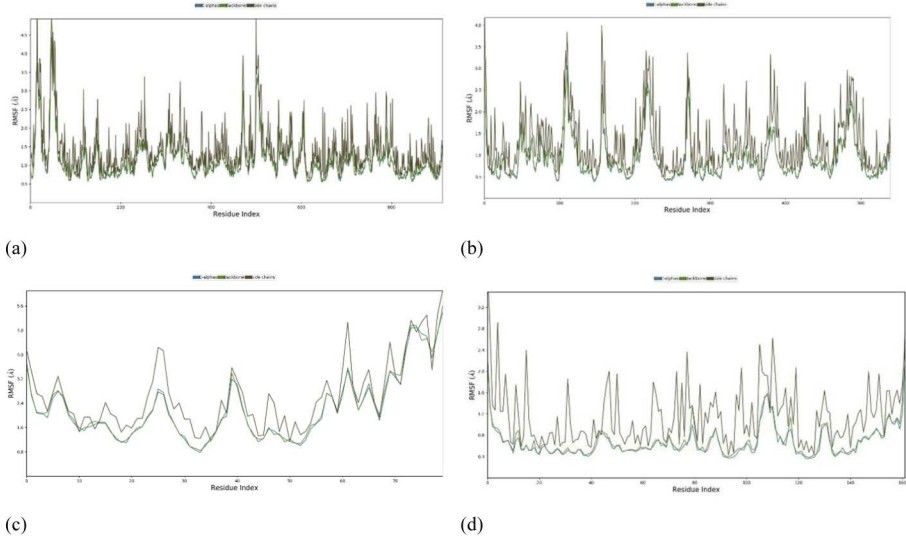

**Fig 12. Root Mean Square Fluctuation with respect of residuesprotein-ligand complexes.** (a) *Phytophthora capsici* (7vem) – Difenoconazole, (b) *Botrytis cinerea* (8H6Q) – Carbendazim, (c) *Fusariumoxysporum* f. sp. *Lycopersici* (8EBB) –Difenoconazole, (d) *Pucciniagraminis* f. sp. *tritic*i (7XDS).

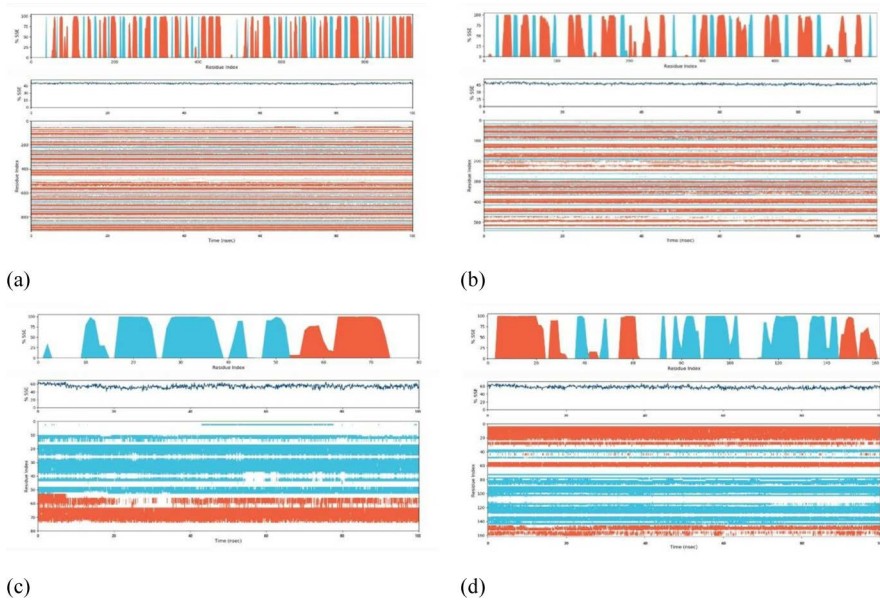

**Fig 13. Protein secondary structure elements (SSE) of protein- ligand complexes.** (a) *Phytophthoracapsici* (7vem) – Difenoconazole, (b) *Botrytis cinerea* (8H6Q)–Carbendazim, (c) *Fusariumoxysporum* f. sp. *Lycopersici* (8EBB) – Difenoconazole, (d) *Pucciniagraminis* f. sp. *tritici* (7XDS).

chain align in an extended conformation and form hydrogen bonds with each other. Loops or turns are regions of the protein structure that do not form regular hydrogen bonding patterns and connect alpha helices and beta strands. Once SSEs are identified, various analyses can be performed to understand their dynamics and interactions. Quantitative measures such as the length of alpha helices or beta strands, the number of hydrogen bonds within SSEs, and the persistence of SSEs over time can be calculated. Changes in SSEs during the simulation,

such as the formation or disruption of helices or strands, can be monitored to study protein folding, unfolding, or conformational changes. SSEs can also be correlated with other structural or dynamic properties, such as solvent accessibility, protein-ligand interactions, or protein-protein interactions. However, SSEs, and SASA mention the stable configuration of docked complexes.

**Protein Ligand interaction by bond.** In MD simulations, the interactions between a protein and a ligand (small molecule) are crucial for understanding the mechanism of binding and the stability of the complex. These interactions can involve various types of bonds, including covalent bonds, hydrogen bonds, hydrophobic interactions, and electrostatic interactions. Covalent bonds between the protein and ligand can form when specific reactive groups are present.

Analysis involves identifying the formation or breaking of covalent bonds over the course of the simulation. These interactions are typically less common in non-covalent protein-ligand interactions but are important in certain types of enzyme-substrate interactions or covalent inhibitors. Hydrogen bonds between the protein and ligand play a significant role in stabilizing the complex. Analysis involves identifying hydrogen bond formation and rupture events. Criteria such as distance and angle cutoffs are used to define hydrogen bonds. The frequency, duration, and strength of hydrogen bonds can be analyzed to understand their contribution to binding affinity attached (Fig 14). Hydrophobic interactions occur between non-polar regions of the ligand and protein. Analysis involves identifying hydrophobic contacts based on the proximity of non-polar atoms. Solvent-accessible surface area (SASA) analysis can also provide insights into the burial of hydrophobic residues upon ligand binding.

**Radius of gyration (Rg) of WT, mutations, Solvent-accessible surface area (SASA).** The distribution of an object's mass around its axis of rotation is measured by the radius of gyration, or Rg. It is frequently used to characterize the spatial distribution of mass in a rigid body or a system of particles in physics and engineering. The radius of gyration for a

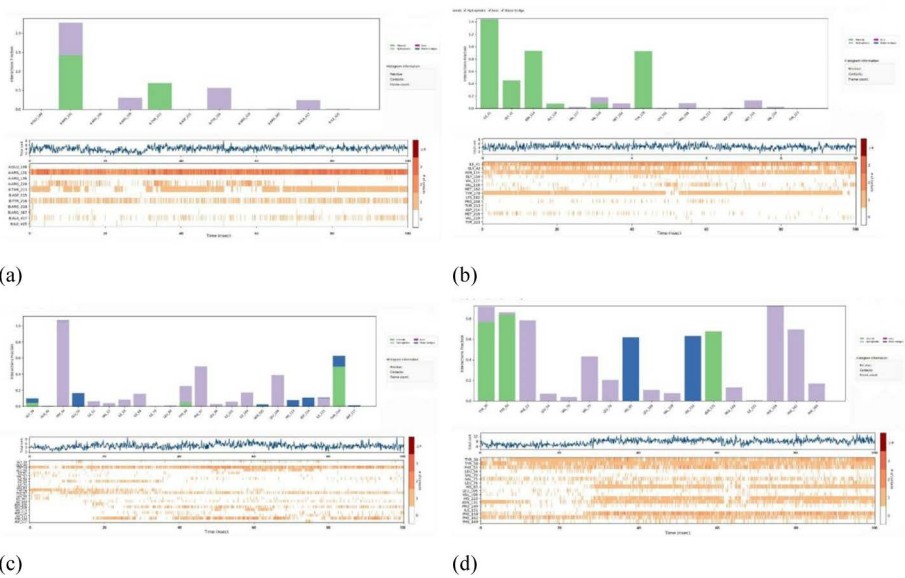

(a)   (b)   (c)   (d)

**Fig 14. Protein Ligand interaction by bond of protein- ligand complexes.** (a) *Phytophthoracapsici* (7vem) – Difenoconazole, (b) *Botrytis cinerea* (8H6Q)–Carbendazim, (c) *Fusariumoxysporum* f. sp. Lycopersici (8EBB) –Difenoconazole, (d) *Pucciniagraminis* f. sp. *tritici* (7XDS).

particular system or object is equal to the square root of the moment of inertia divided by the total mass. In terms of math, it is stated as:

$$Rg = \sqrt{\frac{I}{m}}$$

Here,

Rg is the radius of gyration, I is the moment of inertia of the object or system,m is the total mass.

From the radius of gyration data, it is found that stability and response to external forces under different loading conditions of structure is valid and stable. In MD simulations, the solvent-accessible surface area (SASA) is an essential metric that offers important details on how atoms in a bimolecular system are exposed to the surrounding solvent. SASA is frequently used to investigate how ligands and proteins interact. Variations in the solvent accessibility can reveal areas that are revealed during unbinding or concealed during ligand binding. Finding binding locations and comprehending the energetics of protein-ligand interactions benefit from this. From data of MD simulations, with SASA analysis, it helps to show that the working procedure of docking is valid by experimental data by providing insights into the dynamic behavior of molecules, revealing solvent exposure and flexibility that may not be evident in static structures shown (Fig 15).

**Torsional flexibility in Molecular Docking studies.** Ligands frequently have chemical structures that are flexible, and their torsional flexibility—the capacity to rotate around particular bonds—allows them to take on many conformations. In order for the ligand to locate an energetically advantageous binding position within the protein's binding site, this flexibility is essential. Torsional flexibility is taken into account by the software while simulating the interaction between a ligand and a protein. In order to investigate several conformations and determine the ideal binding posture inside the protein's binding site, the ligand is permitted to rotate around its rotatable links.

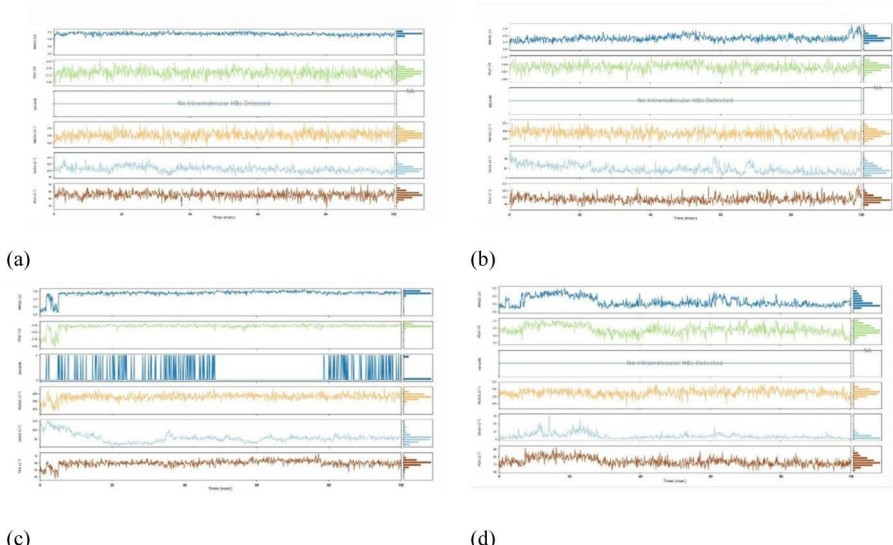

(a)  (b)

(c)  (d)

**Fig 15. Radius of gyration (Rg) of WT, mutations, Solvent-accessible surface area (SASA).** (a) *Phytophthoracapsici* (7vem) – Difenoconazole, (b) *Botrytis cinerea* (8H6Q)–Carbendazim, (c) *Fusariumoxysporum* f. sp. Lycopersici (8EBB) –Difenoconazole, (d) *Pucciniagraminis* f. sp. *tritici* (7XDS).

In MD simulations of protein-ligand interactions, torsion refers to the rotation of specific dihedral angles around covalent bonds within the ligand or the protein. Molecular dynamics simulations involve the computational modeling of the dynamic behavior of molecules over time. Torsional movements are crucial in understanding the conformational changes and flexibility of both the protein and the ligand during their interaction.

First of all, MD simulations track the positions and movements of atoms over time, allowing for the observation of conformational changes. Torsional movements influence the overall flexibility and shape of the protein-ligand complex from the first picture where most of residues are in right shape among 90–180. Secondly, the torsional energy landscape provides insights into the stability and energetically favorable conformations of the protein-ligand complex. All of residues are in lower energy almost zero energy, showing the highest stability having (Fig 16).

## Ramachandran plot for Docked protein complex

The Ramachandran plot's precise characteristics and importance would be described in the pertinent literature or documentation if it is, in fact, a tool or analysis approach used to evaluate the stability of proteins. Studying a variety of elements, including protein folding, thermodynamic characteristics, and interactions with other molecules, is frequently necessary for the stability analysis of proteins. The statistical distribution of the possible combinations of the backbone the dihedral angles υ and ψ is presented in the Ramachandran the following diagram. The Ramachandran plot's permissible regions, in theory, indicate the potential values of the Phi/Psi angles for an amino acid (X) in an ala-X-ala tripeptide. It can be seen that the plots generally favor amino acid resets above 96%, which give the protein ligand complex a double configuration taken from (Fig 17).

## Conclusion

The combination of in silico investigations using AutoDock by PyRx and Molecular Docking by Glide from the Schrödinger suite has yielded valuable insights into potential fungicide candidates against four plant pathogenic fungal proteins -*Phytophthora capsici, Botrytis cinerea, Fusarium oxysporum* f. sp. *lycopersici,* and *Puccinia graminis* f. sp. *tritici.* The robust molecular docking simulations have convincingly validated the pharmacological findings, demonstrating the binding of various fungicide compounds to their respective target proteins. The main

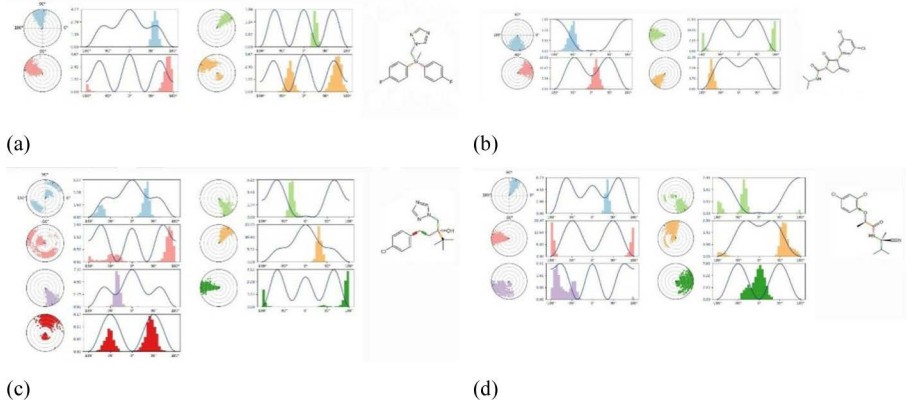

(a)                                                        (b)

(c)                                                        (d)

**Fig 16. Torsional flexibility In Molecular Docking Studies protein- ligand complexes.** protein-ligand complexes (a) *Phytophthoracapsici* (7vem) – Difenoconazole, (b) *Botrytis cinerea* (8H6Q)–Carbendazim, (c) *Fusariumoxysporum* f. sp. *Lycopersici* (8EBB) –Difenoconazole, (d) *Pucciniagraminis* f. sp. *tritici* (7XDS).

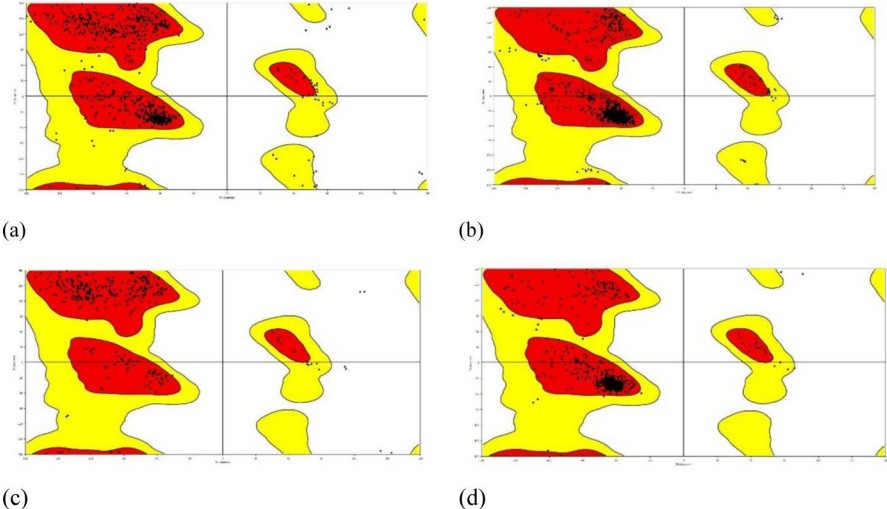

(a)

(b)

(c)

(d)

**Fig 17. Torsional flexibility In Molecular Docking Studies protein- ligand complexes.** (a) *Phytophthoracapsici* (7vem) – Difenoconazole, (b) *Botrytis cinerea* (8H6Q)–Carbendazim, (c) *Fusariumoxysporum* f. sp. Lycopersici (8EBB) –Difenoconazole, (d) *Pucciniagraminis* f. sp. *tritici* (7XDS).

modes of interaction between the compounds and proteins, revealed through these simulations, mostly involve hydrogen and hydrophobic bonds. These interactions play a crucial role in the development of structurally-oriented fungicides, highlighting the importance of understanding the intricate engagement between proteins and ligands. The accepted standard for a fungicide, with docking scores greater than −6.00 kcal/mol, aligns with the results, demonstrating the potential effectiveness of the identified compounds. Among the four fungal pathogens, the fungicides L01, L03, L04, L13, L14, L17, L18, and L19 consistently exhibited significantly higher binding affinities compared to other ligands, with docking scores ranging from −3.00 to −9.20 kcal/mol. In case of DFT, L16 demonstrates the highest electrophilicity (5.174), indicating its strong potential as an electron acceptor, with high chemical stability due to its significant hardness (3.783). Next, L20 shows the highest absorption in the human intestine (1.0000) but has low blood-brain barrier permeability and mixed activity in P-glycoprotein interactions, making it a strong candidate for targeted systemic applications with limited CNS penetration. Ligand L03 demonstrates strong binding affinity against *Phytophthora capsica*, highlighting its potential as an effective fungicide; however, further studies on environmental impact and non-target toxicity are essential for sustainable agricultural use in both AutoDock and Glide simulations. This convergence of results from different molecular docking methodologies emphasizes the reliability and consistency of L03's binding affinity for this particular fungal pathogen.

## Supporting information

**S1 Fig. Optimized structure of molecules.**
(DOCX)

**S2 Fig. Frontier Molecular Orbital of HOMO-LUMO for optimized structure of molecules.**
(DOCX)

**S3 Fig. Electrostatic potential map for optimized structure of molecules.**
(DOCX)

**S4 Fig. Molecular docking pose of molecules.**
(DOCX)

## Acknowledgments

Authors are expressed their thankful to the IUBAT Innovation and Entrepreneurship Center (IIEC), IUBAT—International University of Business Agriculture and Technology, 4-Embankment Drive Road, Sector 10 Uttara Model Town, Dhaka 1230, Bangladesh for their technical supports to carry on this research progress.

## Author contributions

**Conceptualization:** Ajoy Kumer, Swapan Kumar Roy.

**Data curation:** Mollah Naimuzzaman, Md Mahabub Hasan, Abu Yousuf Hossin, Swapan Kumar Roy.

**Formal analysis:** Mollah Naimuzzaman, Md Mahabub Hasan, Abu Yousuf Hossin.

**Funding acquisition:** Ajoy Kumer, Jamal Uddin.

**Investigation:** Mollah Naimuzzaman, Abu Yousuf Hossin.

**Methodology:** Mollah Naimuzzaman, Md Mahabub Hasan, Abu Yousuf Hossin.

**Project administration:** Ajoy Kumer, Mohammad Harun-Ur-Rashid, Swapan Kumar Roy, Abu Noman Faruq Ahmmed, Jamal Uddin.

**Resources:** Mollah Naimuzzaman, Md Mahabub Hasan, Abu Yousuf Hossin, Mohammad Harun-Ur-Rashid, Abu Noman Faruq Ahmmed.

**Software:** Ajoy Kumer, Mohammad Harun-Ur-Rashid.

**Supervision:** Ajoy Kumer, Swapan Kumar Roy, Abu Noman Faruq Ahmmed, Jamal Uddin.

**Validation:** Md Mahabub Hasan, Ajoy Kumer, Abu Yousuf Hossin.

**Visualization:** Mollah Naimuzzaman.

**Writing – original draft:** Ajoy Kumer.

**Writing – review & editing:** Ajoy Kumer, Abu Yousuf Hossin, Mohammad Harun-Ur-Rashid, Swapan Kumar Roy, Abu Noman Faruq Ahmmed, Jamal Uddin.

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
