## [Decision Letter · Decision Letter 0]

8 May 2024

PONE-D-24-06911Inquisition of Novel Targets for Potential Fungicides against four plant pathogenic fungal by in silico, molecular docking, Molecular dynamics and DFTPLOS ONE

Dear Dr. Kumer,

Thank you for submitting your manuscript to PLOS ONE. After careful consideration, we feel that it has merit but does not fully meet PLOS ONE’s publication criteria as it currently stands. Therefore, we invite you to submit a revised version of the manuscript that addresses the points raised during the review process.

**ACADEMIC EDITOR:**

1. Title of the manuscript needs revision. It looks incomplete as "four plant pathogenic fungal".

2. The manuscript lacks discussion on the bio-availability and toxicity of the said fungicides.

3. More data has been demanded by a Reviewer-2 and it is doable to use more in silico tools to elaborate the results especially the impact  of these fungicides on living cells  if these fungicides get enter into the cells.

4. Results lack description keeping in view the biological systems that may come in contact with these molecules..

We look forward to receiving your revised manuscript.

Kind regards,

Faiz Ahmad Joyia, Ph.D.

Academic Editor

PLOS ONE

Journal Requirements:

[N/A]. 

5. We note that you have indicated that there are restrictions to data sharing for this study. PLOS only allows data to be available upon request if there are legal or ethical restrictions on sharing data publicly. For more information on unacceptable data access restrictions, please see http://journals.plos.org/plosone/s/data-availability#loc-unacceptable-data-access-restrictions. 

Additional Editor Comments:

Comments by Reviewer 2.

Author should add more in-silico data from multiple software and tools to evaluate these findings. This is very limited data for such important fungal pathogens.

Reviewers' comments:

Reviewer's Responses to Questions

**Comments to the Author**

1. Is the manuscript technically sound, and do the data support the conclusions?

Reviewer #1: Yes

Reviewer #2: No

2. Has the statistical analysis been performed appropriately and rigorously?

Reviewer #1: I Don't Know

Reviewer #2: N/A

3. Have the authors made all data underlying the findings in their manuscript fully available?

Reviewer #1: Yes

Reviewer #2: Yes

4. Is the manuscript presented in an intelligible fashion and written in standard English?

Reviewer #1: Yes

Reviewer #2: No

5. Review Comments to the Author

Reviewer #1: The review for the manuscript entitled “Inquisition of Novel Targets for Potential Fungicides against four plant pathogenic fungal by in silico, molecular docking, Molecular dynamics and DFT”.

Authors have explored potential candidates for fungicides against 4 fungal proteins. They have used in sillico, molecular docking simulations, and molecular dynamic simulation for selecting the nature inhibitors against fungal pathogens.

In opinion:

The manuscript title succinctly reflects the research's core. It's a meticulously prepared document:

The introduction furnishes a thorough overview, laying the groundwork for the study.

The methods section meticulously explains the procedures, ensuring reproducibility for fellow researchers.

Results and figures are clearly presented, facilitating comprehension of the findings.

The discussion and conclusion sections adeptly analyze the results, offering valuable insights into the study's implications.

Reviewer #2: Author should add more in-silico data from multiple software and tools to evaluate these findings. This is very limited data for such important fungal pathogens. Author should start with one fungal pathogen with the fungicide first.

6. PLOS authors have the option to publish the peer review history of their article (what does this mean? ). If published, this will include your full peer review and any attached files.

**Do you want your identity to be public for this peer review?** For information about this choice, including consent withdrawal, please see our Privacy Policy .

Reviewer #1: **Yes: ** Subhash Chandra

Reviewer #2: No

---

## [Decision Letter · Decision Letter 1]

18 Nov 2024

PONE-D-24-06911R1Inquisition of Novel Fungicidal Targets for Combating Root Rot, Gray Mold, Fusarium Wilt, and Cereal Rust through In Silico Techniques, Molecular Docking, Dynamics, and DFT AnalysisPLOS ONE

Dear Dr. Kumer, 

Thank you for submitting your manuscript to PLOS ONE. After careful consideration, we feel that it has merit but does not fully meet PLOS ONE’s publication criteria as it currently stands. Therefore, we invite you to submit a revised version of the manuscript that addresses the points raised during the review process.

Revised version of the manuscript is satisfactory. The authors are requested to perform minor revisions as suggested by the authors.

We look forward to receiving your revised manuscript.

Kind regards,

Faiz Ahmad Joyia, Ph.D.

Academic Editor

PLOS ONE

Journal Requirements:

Additional Editor Comments:

The revised version of manuscript is satisfactory. The authors are requested to perform minor revision as suggested by the reviewers.

Reviewers' comments:

Reviewer's Responses to Questions

**Comments to the Author**

1. If the authors have adequately addressed your comments raised in a previous round of review and you feel that this manuscript is now acceptable for publication, you may indicate that here to bypass the “Comments to the Author” section, enter your conflict of interest statement in the “Confidential to Editor” section, and submit your "Accept" recommendation.

Reviewer #2: (No Response)

Reviewer #3: (No Response)

2. Is the manuscript technically sound, and do the data support the conclusions?

Reviewer #2: Yes

Reviewer #3: (No Response)

3. Has the statistical analysis been performed appropriately and rigorously?

Reviewer #2: N/A

Reviewer #3: (No Response)

4. Have the authors made all data underlying the findings in their manuscript fully available?

Reviewer #2: Yes

Reviewer #3: (No Response)

5. Is the manuscript presented in an intelligible fashion and written in standard English?

Reviewer #2: No

Reviewer #3: (No Response)

6. Review Comments to the Author

Reviewer #2: This revised draft still needs formatting and grammatical correction throughout the draft. For example, Figure 6 to 9 legend font size is different. Give brief description for all figure legends. Title under each figure does not convey any information.

Cite the in-silico reference https://doi.org/10.55730/1300-011x.3191 with REF 9 for the statement “particularly in silico techniques like molecular docking[9]”

Do not use abbreviations in figure legend. For example, figure 2 legend, HOMO and LUMO. Write full name.

Check the formatting for the heading “A comparative study for docking Results”.

Reviewer #3: (No Response)

7. PLOS authors have the option to publish the peer review history of their article (what does this mean? ). If published, this will include your full peer review and any attached files.

**Do you want your identity to be public for this peer review?** For information about this choice, including consent withdrawal, please see our Privacy Policy .

Reviewer #2: **Yes: ** Hira Kamal

Reviewer #3: No

---

## [Author Response · Author response to Decision Letter 1]

10 Dec 2024

The paper presents a comprehensive computational study on the identification of potential fungicides against four fungal proteins. The authors employ a range of computational methods, including molecular docking, molecular dynamics simulations, and pharmacokinetics studies. However here are some suggestions

1. Title should be shortened and more concise.

Answer: It has revised and have made short as

“Computational and In silico study of Novel Fungicides against Combating Root Rot, Gray Mold, Fusarium Wilt, and Cereal Rust.”

2. Should add relevant keywords that accurately reflect the content of the paper.

Answer: It has revised and added as

Density Functional Theory, HOMO, LUMO, Molecular docking, ADMET and Molecular Dynamics

3. Don’t use jargon in title

Answer: It has revised and sincerely removed.

4. Add description or full form of these proteins i.e. 7VEM, 8H6Q, 8EBB, 7XDS, at

least at first time use.

Answer: It has revised and added as:

The proteins 7VEM, 8H6Q, 8EBB, and 7XDS, derived from various devastating plant pathogens, represent critical targets for computational drug discovery aimed at combating agricultural diseases. The protein 7VEM, identified as NADPH-assisted quinone oxidoreductase from Phytophthora capsici, plays a crucial role in maintaining redox balance essential for the pathogen's survival and virulence, making it a prime target for inhibitor design. Solved at 2.39 Å resolution with an R-work value of 0.167, its structural fidelity and lack of mutations ensure reliable insights for virtual screening and molecular docking. Similarly, the Class I sesquiterpene synthase BCBOT2 (apo) from Botrytis cinerea (PDB ID: 8H6Q), resolved at 2.00 Å with R-free and R-work values of 0.195 and 0.168 respectively, is vital for sesquiterpene biosynthesis linked to the pathogen's virulence. Its structural data offer a robust framework for identifying small molecules to disrupt sesquiterpene synthesis, attenuating B. cinerea's pathogenicity. The SIX6 protein from Fusarium oxysporum f. sp. lycopersici (PDB ID: 8EBB), despite its classification as a protein of unknown function, plays a critical role in the virulence mechanism of this vascular wilt pathogen, which devastates crops like tomatoes. Its high-resolution structure at 1.88 Å, with R-free and R-work values of 0.221 and 0.193, supports molecular docking, virtual screening, and lead optimization to design antifungal agents. Finally, the AvrSr35 effector protein from Puccinia graminis f. sp. tritici (PDB ID: 7XDS), resolved at 2.06 Å with reliable R-values (R-free: 0.287, R-work: 0.257, Observed: 0.258), provides critical insights into wheat stem rust pathogenicity. Its potential as a target for structure-based drug design is underscored by its well-defined active sites and suitability for molecular dynamics simulations and virtual screening. Together, these structural models offer a comprehensive platform for advancing the design of inhibitors to mitigate plant pathogen-induced diseases Table 1.

Table 1. Protein information.

Title PDB ID:7VEM PDB ID:8H6Q PDB ID:8EBB PDB ID:7XDS

Organism Phytophthora capsici Botrytis cinerea Fusarium oxysporum f. sp.lycopersici Puccinia graminis f. sp. tritici

Resolution 2.39 Å 2.00 Å 1.88 Å 2.06 Å

R-Value Free 0.224 0.195 0.221 0.287

Ramachoron plot, % 89.5 93.8 91.8 94.7

5. Also add sequence characterization study for the given proteins

Answer: It has revised and added which is almost relevant answer of question 4.

6. Add details of figure in legends

Answer: Answer: It has revised and added as:

The study of a computational procedure to determine the quantum calculations of any chemical species requires the optimization of the molecular structure, which is an important aspect of its structural geometry.

Additionally, accurate computational parameters are obtained by determining the most stable configuration of any chemical structure. In this study, all compounds underwent computational optimization using the DFT functional, and their primary and most stable configuration was observed with minimal energy required for optimization. The antifungal ligands are Azoxystrobin(L01), Cyproconazole(L02), Difenoconazole (L03), Tebuconazole (L04), Tricyclazole (L05), Chlorothalonil (L06), Benalaxyl (L07),Bismerthiazol (L08), Carbendazim (L09), Hexaconazole (L10), Thiram (L11), Carboxin (L12), Iprodione (L13), Kresoxim-Methyl (L14), Cymoxanil (L15), Dichloran (L16), Propiconazole (L17), Dimethomorph (L18), Pyraclostrobin (L19) andAmetoctradin (L20) are shown in Fig 1 and supplementary 1.

L01 L02

L03 L04

---

## [Editor Report · Decision Letter 2]

15 Dec 2024

Computational and In silico study of Novel Fungicides against Combating Root Rot, Gray Mold, Fusarium Wilt, and Cereal Rust

PONE-D-24-06911R2

Dear Dr. Ajoy Kumer,

We’re pleased to inform you that your manuscript has been judged scientifically suitable for publication and will be formally accepted for publication once it meets all outstanding technical requirements.

Kind regards,

Faiz Ahmad Joyia, Ph.D.

Academic Editor

PLOS ONE
---

## [Editor Report · Acceptance letter]

PONE-D-24-06911R2

PLOS ONE

Dear Dr. Kumer,

I'm pleased to inform you that your manuscript has been deemed suitable for publication in PLOS ONE. Congratulations! Your manuscript is now being handed over to our production team.

Kind regards,

on behalf of

Dr. Faiz Ahmad Joyia

Academic Editor

PLOS ONE